# Slippery damper of an overlay for arresting and manipulating droplets on nonwetting surfaces

Xing Han[1,2], Wei Li[1,2], Haibo Zhao[1,2,3], Jiaqian Li[1,2], Xin Tang 🔟 [1,2 ✉] & Liqiu Wang 🔟 [1,2 ✉]

In diverse processes, such as fertilization, insecticides, and cooling, liquid delivery is compromised by the super-repellency of receiving surfaces, including super-hydro-/omni-phobic and superheated types, a consequence of intercalated air pockets or vapor cushions that promote droplet rebounds as floating mass-spring systems. By simply overlaying impacting droplets with a tiny amount of lubricant (less than 0.1 vol% of the droplet), their interfacial properties are modified in such a way that damper-roller support is attached to the mass-spring system. The overlayers suppress the out-of-plane rebounds by slowing the departing droplets through viscous dissipation and sustain the droplets' in-plane mobility through self-lubrication, a preferential state for scenarios such as shedding of liquid in spray cooling and repositioning of droplets in printing. The footprint of our method can be made to be minimal, circumventing surface contamination and toxification. Our method enables multifunctional and dynamic control of droplets that impact different types of nonwetting surfaces.

[1] Department of Mechanical Engineering, The University of Hong Kong, Hong Kong, Hong Kong. [2] Zhejiang Institute of Research and Innovation, The University of Hong Kong, Hangzhou, Zhejiang, China. [3] Department of Mechanics and Aerospace Engineering, Southern University of Science and Technology, Shenzhen, Guangdong, China. ✉email: tangxin@connect.hku.hk; lqwang@hku.hk

Enhancing liquid deposition is fundamental to agricultural sprays, insecticides, spray cooling, droplet-based printing, cosmetics and many other applications[1–4]. On nonwetting surfaces, including superhydrophobic[5–13], superomniphobic[14,15] and superheated surfaces (a superheated surface is defined as a substrate whose temperature is above the droplet's Leidenfrost temperature)[16,17], impacting droplets rebound in ~10 ms by retrieving their kinetic energy through shape restoration, behaving like a floating mass-spring system, which is a consequence of the intercalated air pockets or vapour cushions that allow liquid levitations[18–23]. This repellency compromises liquid delivery on crop leaves, insect cuticles and overheated workpieces, causing issues such as pesticide overuse, soil contamination and water resource waste.

Additives have been dissolved in water to promote its deposition on superhydrophobic surfaces. Through real-time surface tension reduction[24,25] or in situ pinning defect patterning[2,26,27], additives immobilize impacting droplets, disabling both out-of-plane and in-plane mobility as though a fixed support is provided to the mass-spring system. In many other applications, post-deposition liquid manipulation can be as crucial as enhanced liquid retention. For example, reliable coolant depositions on hot surfaces streamline spray cooling. The heat transfer can be further expedited by increasing the coolant turnover rate if heated liquids are frequently shed off. A method that suppresses out-of-plane liquid escape and enables in-plane liquid control is thus highly beneficial.

Here we present a slippery damper strategy to enhance liquid deposition and allow post-deposition manipulations by capping the millimetre-size impacting droplets with micro-/nano-thick liquid overlayers. The overlayer is introduced using a co-flow microfluidic device, which is a tool that enables us to precisely tune the liquid thickness. By fulfilling the spreading criterion, a tiny amount of immiscible liquid homogeneously overlays the droplet, substantially mediating its interfacial properties. To prevent the out-of-plane rebounds, the overlayer spontaneously collapses the air or vapour layer, establishing liquid/solid contacts. As the droplet retracts, the pinned overlayer drains its kinetic energy through viscous dissipation, functioning as a damper to slow the departing droplets. Such a droplet-arresting capability can effectively suppress enhanced droplet rebounds on wired or curved repellent surfaces and yield a fourfold-higher spray cooling rate on superheated substrates. To maintain in-plane mobility, the overlayer lubricates and decreases the liquid/solid friction, offering roller support to the droplets. When the overlayer amount is sufficient to immerse the substrate asperities, the overlaid droplets can smoothly slide after deposition, as though they carry their own lubricant to turn the substrates in situ into slippery liquid-infused surfaces. Using field-responsive fluids, such as ferrofluids, the overlayer acts as the force mediator, allowing active droplet control. The overlayer strategy provides enhanced liquid retention that permits multifunctional liquid controls. Such reconcilement of the two seemingly contradictory capabilities has great potential in a wide range of applications, such as bioprinting, chemical handling (deposition of hazardous solutions) and spray cooling.

## Results

**Enhanced deposition.** To verify our strategy (Fig. 1a, b), we fabricate three nonwetting surfaces as models. As shown in Fig. 1c, for the superhydrophobic type, the silicon wafer is spray-coated with silanized silica nanoparticles (see "Methods" for details). After baking, nanoscale porous textures form, providing a static water contact angle of $152.0 \pm 1.3°$. Similarly, the superomniphobic surface is fabricated by coating the silicon wafer with

a polysiloxane/silica stock suspension[28]. The treated surface repels hexadecane with a contact angle of $151.2 \pm 1.5°$. For the superheated surface, we heated an aluminium plate to 251 °C. Water atop it becomes Leidenfrost droplets that are thermally insulated by levitating vapour cushions.

By simply fulfilling the liquid spreading criterion $S > 0$ ($S = \gamma_d - \gamma_o - \gamma_{d/o}$, where $S$ is the spreading coefficient, $\gamma$ represents surface/interfacial tension, subscripts d, o and d/o denote droplet, overlayer and droplet/overlayer interface, respectively), the preferentially wetting liquid automatically coats the droplet (Supplementary Table 1)[29,30]. The droplet is overlaid using a co-flow microfluidic device[31], whereby the flow rates of two phases, $q_o$ and $q_d$, are precisely tuned through syringe pumps whose minimum feed rate is 1 nl min$^{-1}$ and accuracy is $\pm 0.5\%$ (see "Methods" for details, Supplementary Fig. 1a and Supplementary Movie 1). The equivalent overlayer thickness $c$ is calculated as $c = 0.5 D_0 (1 + q_o/q_d)^{1/3} - 0.5 D_0$, due to the conservation of mass[32–34] $q_d/(q_d + q_o) = D_0^3/(D_0 + 2c)^3$, where $D_0$ is the diameter of the inner droplet. Alternative methods such as counterspraying or sliding atop the overlayer liquid (Supplementary Fig. 1b, c) can coat droplets with high throughput. The ultrathin overlayer modulates the droplet's interfacial properties, as the overlaid ones now appear to be partially wetting on the modelled surfaces (Fig. 1c).

To examine the dynamic performance of our strategy, we liberated a water droplet ($D_0 = 2.0$ mm) overlaid with 60-μm-thick silicone oil from a height of 26 mm onto the superhydrophobic surface (Fig. 1d, Supplementary Fig. 2 and Supplementary Movie 2). Unlike the bouncing pure water droplet, the capped droplet rapidly impregnates the solid upon contact, instantaneously breaking the superhydrophobicity. During the retraction, the overlayer anchors its maximum contact length, restraining the droplet rebound. Similarly, as shown in Fig. 1e, we overlay a hexadecane droplet ($D_0 = 1.9$ mm) with 48-μm-thick lubricate oil (Krytox VPF1506) and let it impact the superomniphobic surface at a velocity of 0.59 m s$^{-1}$. Aided by the wetting overlayer, the alkane droplet is captured without rebound (Supplementary Movie 3). To deposit on the superheated surface, the impacting water droplet is coated with silicone oil with a high boiling temperature $T_{boiling}$ ($T_{boiling} > 300$ °C) (Fig. 1f and Supplementary Movie 4). The overlayer stably contacts the heated surface and prevents instantaneous water vapour formation. In this way, the water droplet thermally bridges the underlying hot solids until further droplet bursting is caused by violent boiling.

**Deposition dynamics.** We use high-speed photography up to 5000 frames per second to capture the impacts, enabling us to delineate the role played by the overlayer. Figure 2a shows the evolution of the normalized contact lengths $D(t)/D_0$. Compared with pure water droplets, overlaid droplets take slightly longer to attain the maximum contact length, showing the influence of the overlayer on the impact, as described in previous studies[35–37]. In receding, $D(t)$ of pure water decreases and returns to 0 upon rebound. Regardless of the shape recovery, $D(t)$ is maintained for overlaid ones due to the impregnation of the lubricant. The corresponding Weber number for a droplet is defined as $We = \rho_d D_0 U_i^2/\gamma_k$, with subscript k being d for a pure droplet and d/o for an overlaid droplet, where $\rho_d$ and $U_i$ are the inner droplet density and impact velocity, respectively. As shown in Fig. 2b, by increasing the Weber number, the normalized impact time $t_i$, defined as the time until the arrested droplet reaches the maximum out-of-plane height $H_{max}$, slightly fluctuates but has distinct values for different overlayer viscosities, suggesting a prominent role played by the overlayer.

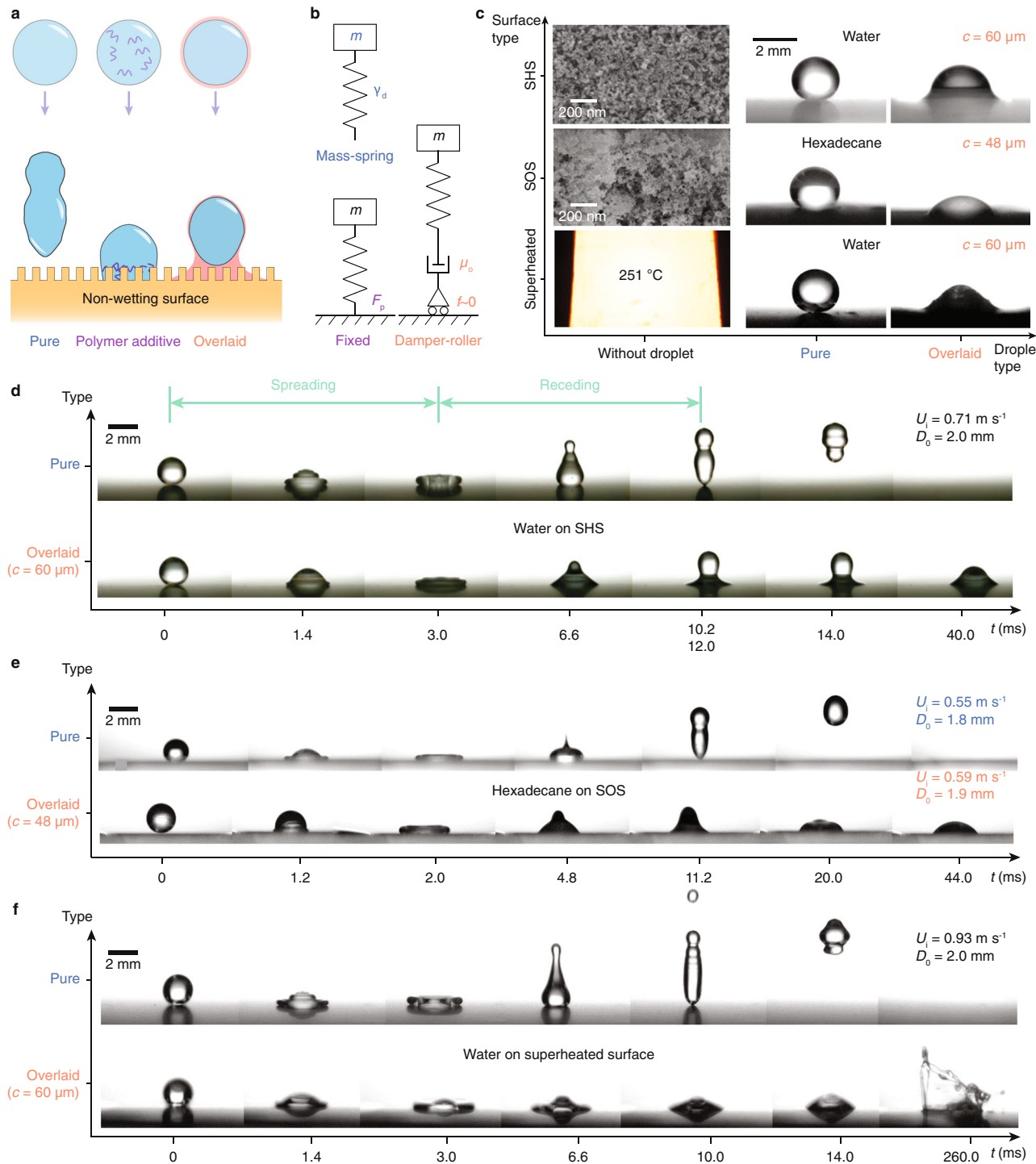

**Fig. 1 Enhanced deposition. a** Schematics showing the liquid overlayer and conventional polymer additives in breaking the super-repellency. **b** Schematics showing the mechanical analogies of the systems in (**a**). Pure water behaves like a floating mass spring because of the surface tension $\gamma_d$; the polymer additives pin the droplet, offering a fixed support, and the overlayer dissipates the kinetic energy (damper) and enables in-plane mobility (roller support). **c** Scanning electron microscopy (SEM) images showing nanotextures of the modelled superhydrophobic surface (SHS) and superomniphobic surface (SOS), and an infrared thermal image showing the superheated surface above the Leidenfrost temperature of water (left column). Microscopy images showing the high water contact angles on the superhydrophobic/superheated surface and the high hexadecane contact angle on the superomniphobic surface (middle column). Microscopy images showing the static states of overlaid droplets on the modelled super-repellent surfaces (right column). Silicone oil (20 mPa s) and lubricant oil (Krytox VPF1506) are chosen as overlayers for water and hexadecane, respectively. Sequential images showing the dynamic impacts of (**d**) water droplets on the superhydrophobic surface ($We_{pure} = 14.0$, $We_{overlaid} = 25.2$), **e** hexadecane droplets on the superomniphobic surface ($We_{pure} = 15.5$, $We_{overlaid} = 72.0$) and **f** water droplets on the superheated surface ($We_{pure} = 24.2$, $We_{overlaid} = 43.6$).

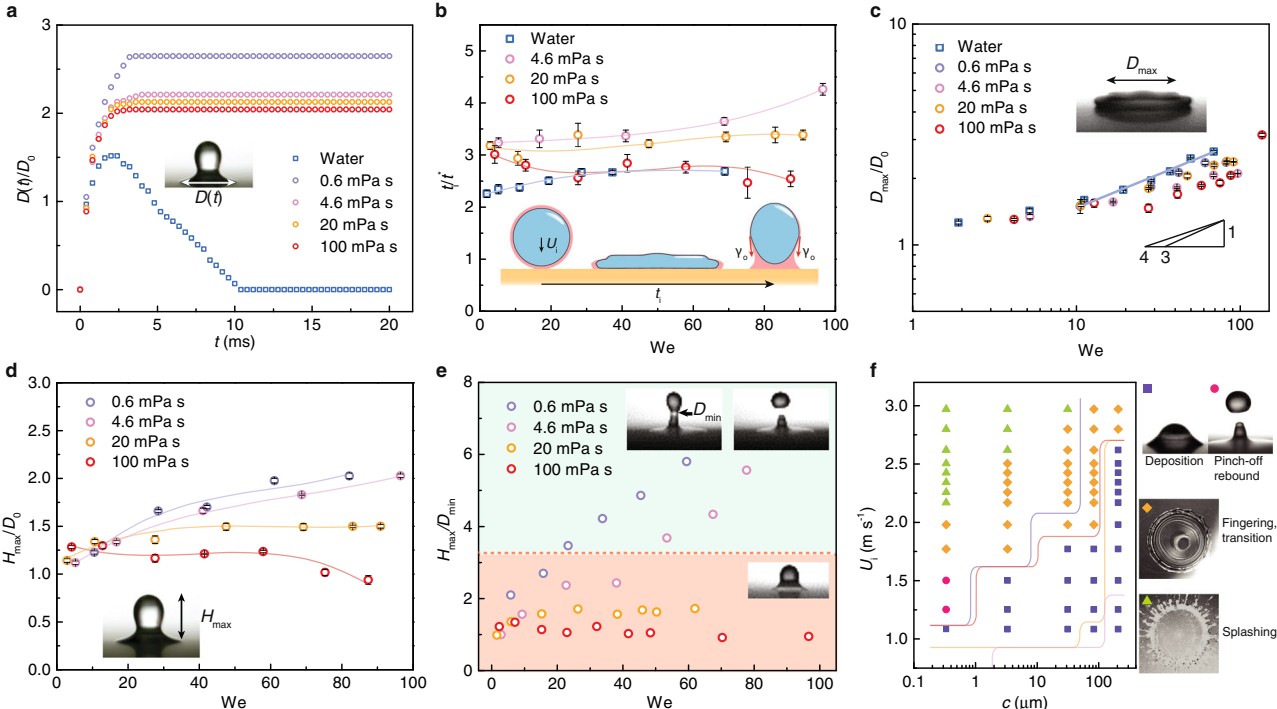

**Fig. 2 Deposition dynamics. a** The temporal evolution of the normalized contact lengths of impacting droplets ($D_0 = 2.0$ mm, $U_i = 0.7$ m s$^{-1}$, $c = 60$ μm). Silicone oils are used as overlayers. **b** The normalized impact time appears to be weakly dependent upon the Weber number and decreases as the overlayer viscosity increases. $t^*$ is the characteristic contact time, $t^* = \sqrt{\rho R_0^3 / \gamma_k}$, where $R_0$ is the droplet radius and subscript k is d for pure water and d/o for overlaid water. Solid lines are drawn to aid the visualization. Insets are schematics showing the spreading and receding of an impacting droplet. Error bars denote the standard deviation over five experiments. **c** The normalized maximum spreading diameters share a similar slope but decrease as the viscosity increases for We >10. The purple line is the linear fitting line for pure water with a slope of 0.28. Error bars denote the standard deviation over five experiments. **d** The normalized maximum out-of-plane height as a function of the Weber number. Solid lines are drawn to aid the visualization. Error bars denote the standard deviation over five experiments. **e** The maximum aspect ratio of droplets after receding as a function of the Weber number. The green and red regions denote deposited and pinch-off rebounding droplets, respectively. The red dashed line denotes the threshold value above which pinch-off occurs. $c = 59$ μm for (**b–e**). **f** Phase diagram of a water droplet overlaid with 100-mPa s silicone oil. Markers show the experimental outcome after impact (purple squares: deposition, magenta circles: pinch-off rebound, orange diamonds: fingering, transition and green triangles: splashing) as a function of the impact velocity and overlayer thickness. Purple, red, orange and pink solid lines denote the boundary of deposition for the overlayer with viscosities of 500, 100, 20 and 4.6 mPa s, respectively. Phase diagrams of water droplets overlaid with 500, 20 and 4.6-mPa s silicone oils are shown in Supplementary Fig. 6.

Interestingly, it has been reported that the addition of the lubricant can actually promote rebound by masking droplet/substrate pinning[37], a result that seems contradictory to our observation. To reconcile the apparent conflict, we compare the impact outcomes of different overlayer viscosities in Supplementary Fig. 3. For a low viscosity (4.6 mPa s), droplets indeed rebound on hydrophilic as well as superhydrophobic glass, as described in the previous study. In contrast, a viscous overlayer (500 mPa s) arrests the droplet, regardless of the same experimental parameters. A bottom view is also obtained by focusing on the top surface of the glass, and we show the subjacent lubricant film state in Supplementary Fig. 4. At a moderate velocity ($U_i = 1.63$ m s$^{-1}$), lubricant spreads and intercalates between the droplet and substrate, eliminating contacts between the two. At a high velocity ($U_i = 3.2$ m s$^{-1}$), the droplet perforates the lubricant film on hydrophilic glass, which is evidenced by the water/glass contact line. However, on superhydrophobic glass no obvious water/substrate contact can be observed, regardless of the intense splashing. Therefore, for enhanced droplet deposition, the subjacent lubricant film is well maintained for the entire impact, and the overlayer plays two roles, namely, draining the droplet energy through viscous dissipation $E_\mu$[38] and absorbing the droplet energy into its surface energy via the work $W \sim \gamma_o D_0^2$ (where $\gamma_o$ is

the overlayer surface tension) done during its pinning to prevent rebound. Provided the restored kinetic energy after viscous draining is less than $W$, then the droplet is arrested.

We then experimentally examine the effect of the overlayer viscosity on the impacts. As shown in Fig. 2c, the normalized maximum spreading diameter $D_{max}/D_0$, where $D_{max}$ is defined as the maximum diameter of expanding water lamella[36], shares a slope similar to that of pure water, whose value is approximately 0.28 for We > 10[39], which is in agreement with the previous observation of compound droplets impacting the hydrophilic surface by Liu and Tran[36]. The addition of an overlayer decreases $D_{max}/D_0$, suggesting less stored surface energy because of the viscous dissipation. We then examine $H_{max}$ at the end of impact as a function of the Weber number (Fig. 2d). Note that $H_{max}$ corresponds to liquid escape in light of the fact that once the aspect ratio $H_{max}/D_{min}$, where $D_{min}$ is the diameter of the rebounding liquid neck, surpasses the threshold value (~3.2)[40,41], the filament pinches off, fleeing away a satellite droplet (Fig. 2e). The threshold value is defined by Rayleigh–Plateau instability and is invariant with respect to the overlayer viscosity. As shown in Fig. 2d, for droplets covered in overlayers of low and moderate viscosity (0.6 and 4.6 mPa s), $H_{max}/D_0$ increases with an increasing Weber number, as expected. However, when the

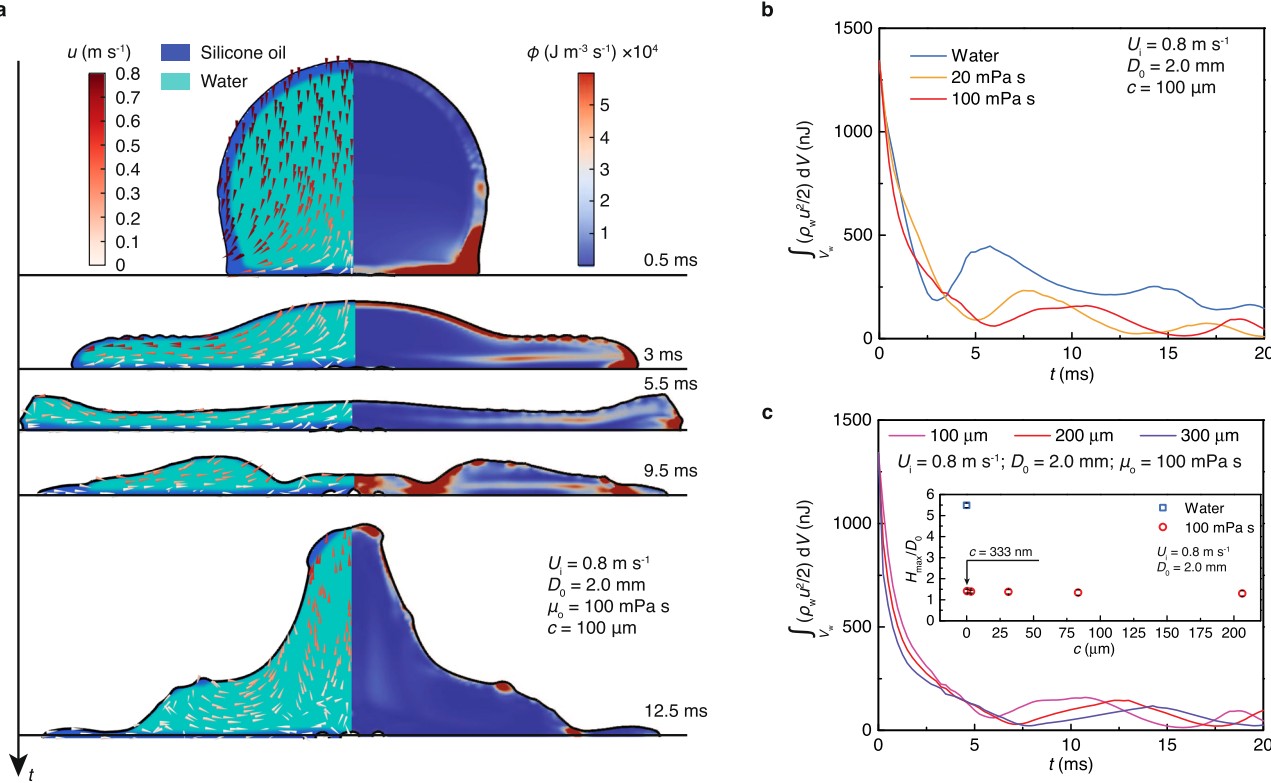

**Fig. 3 Viscous damping. a** Numerically calculated phase, flow field and distribution of the viscous dissipation rate of an impacting overlaid droplet. Viscous dissipation centralises in the overlayer and abounds in the spreading stage. Numerically calculated evolution of the kinetic energy of the water core during the impact for different (**b**) viscosities and (**c**) thicknesses of the overlayer. The inset in (**c**) shows that the experimental maximum out-of-plane height appears nearly independent of thickness for the range of $0.333 < c < 206\,\mu m$, and the error bars denote the standard deviation over three experiments. The overlayer effectively suppresses the rebound as its thickness is as thin as $c \sim 333\,nm$.

overlayer is viscous (20 and 100 mPa s), the $H_{max}/D_0$ plateaus as the Weber number increases. Alternatively, $H_{max}$ can also be reduced by partitioning the inner droplet (Supplementary Fig. 5).

We then empirically summarize the impact outcome with varying overlayer viscosities, overlayer thicknesses, and impact velocities (Fig. 2f and Supplementary Fig. 6) and record four possible impact behaviours: deposition (square), pinch-off rebound (circle), fingering transition (diamond) and splashing (triangle). The splashing of compound droplets on hydrophilic surfaces has been reported by Liu and Tran[35]. In our case, by increasing the viscosity of the ultrathin overlayer, the parameterization region of deposition substantially broadens. Provided silicone oils have approximately the same surface tension and thereby the same work of pinning $W$, we numerically study the distinct viscous dissipation $E_\mu$ within those ultrathin overlayers.

The numerical study of the impact enables us to precisely visualize the flow field and the distribution of the viscous dissipation rate in the overlaid droplet (Supplementary Fig. 7 and Supplementary Note 1). As shown in Fig. 3a, when the droplet contacts the substrate at 0.5 ms, viscous dissipation centralizes in the advancing overlayer front and then redistributes within the top overlayer as the droplet flattens at 3 ms. Viscous dissipation appears stronger in the spreading stage than in the receding stage. The numerically calculated evolution of the kinetic energy of the inner water core in Fig. 3b shows that the addition of a 100-mPa s overlayer can reduce the restored water kinetic energy by more than half. However, by increasing the thickness from 100 to 300 μm, the restored water kinetic energy only decreases by a small amount, and the distribution of viscous dissipation remains similar (Fig. 3c and Supplementary Fig. 8). Such a weak dependence is confirmed by the experiment in which $H_{max}/D_0$

only slightly increases as the overlayer thickness changes from 0.333 to 206 μm (Fig. 3c). The numerically calculated $H_{max}/D_0$ is also similar for 100- and 300-μm-thick overlayers (Supplementary Fig. 8). Therefore, in the experimental range, the overlayer thickness plays a marginal role in the droplet-enhanced deposition.

**Counteract topological perturbations**. Topological perturbations, such as macrotextures and curvatures, are omnipresent in natural/artificial nonwetting surfaces and can promote rebound through momentum redistribution[9,25,42,43]. As shown in Fig. 4a, we pattern a nonwetting wire with a diameter of 100 μm on the superhydrophobic surface. The wire expedites rebounds, reducing the liquid/solid contact time. When the water droplet is overlaid with silicone oil, the wire can still partition the droplet. However, the out-of-plane bouncing tendency is inhibited, and the liquid fragments only laterally move (Supplementary Movie 5). In Fig. 4c, we establish a regime map for droplets impacting macrotextures. As long as the droplet is overlaid, the out-of-plane bouncing is entirely suppressed, even at We = 131.0. Only when the Weber number surpasses a critical value do the droplets fragment into portions. Similar rebounding promotion has also been reported on curved surfaces[43]. As shown in Fig. 4b, we fabricate a super-hydrophobic surface with a curvature of 0.1 mm$^{-1}$. The contact time of the impacting water droplet (We = 32.5) has a ~40% reduction. In stark contrast, the overlaid droplet attaches to the surface without any out-of-plane bouncing (Supplementary Movie 6). The overlayer strategy enables reliable liquid deposition on the complex surface profiles.

It is well known that on surfaces whose temperatures are higher than the Leidenfrost temperature[19] that droplet vapour

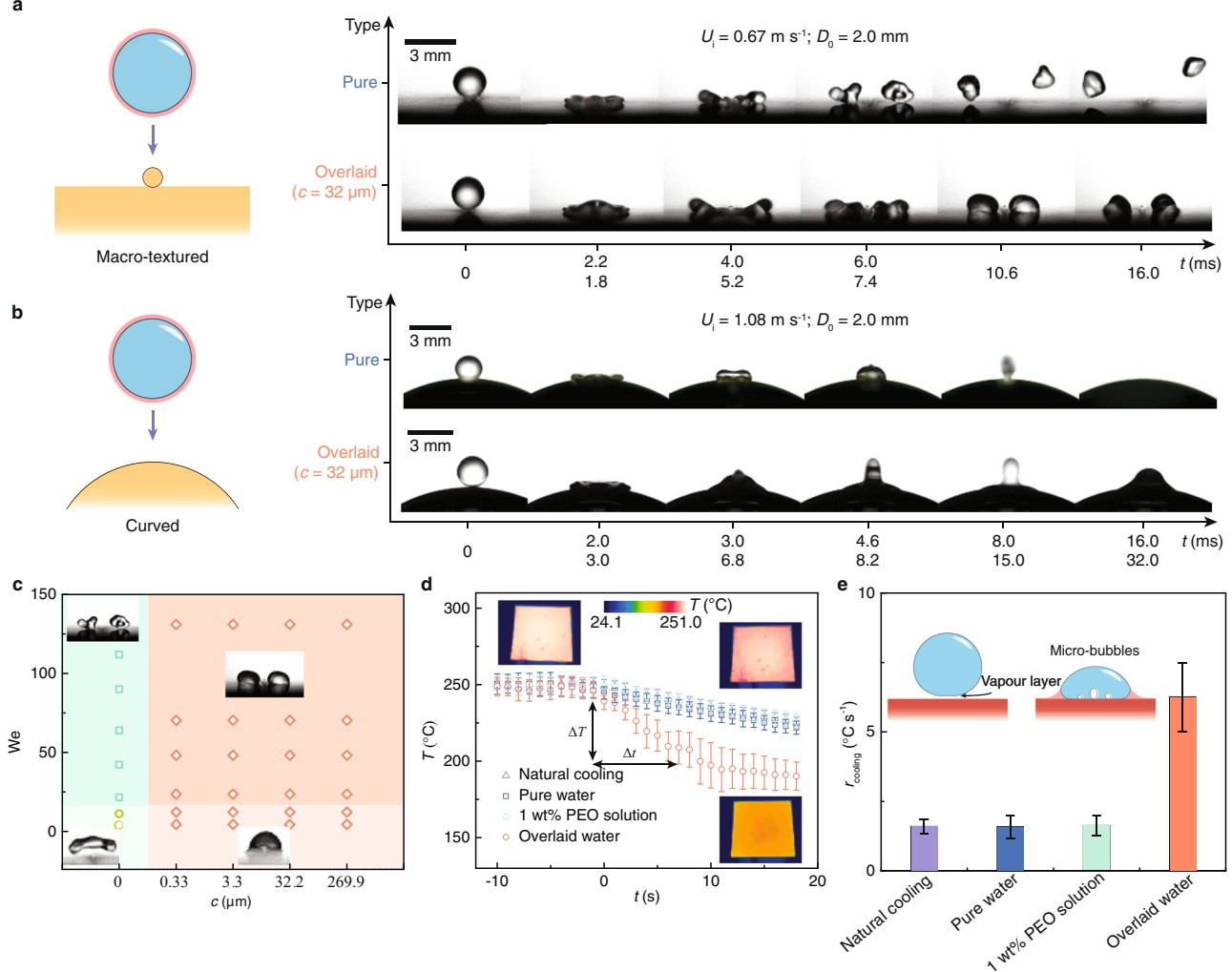

**Fig. 4 Counteract topological perturbations.** Sequential images showing dynamic impacts of water droplets on (**a**) macrotextured (We$_{pure}$ = 12.5, We$_{overlaid}$ = 22.5) and (**b**) curved (We$_{pure}$ = 32.5, We$_{overlaid}$ = 58.5) superhydrophobic surfaces. The overlayers thwart the two rebound upgrading mechanisms. **c** The phase diagram maps the impacting outcomes of the scenario in (**a**). The insets are the microscope images showing droplets after impacts in different phases. The green and red regions denote rebounding and deposited states, respectively. **d** Temperature evolution of superheated aluminium plates using different water cooling strategies. The overlaid water expedites the cooling. The insets are infrared thermal images of aluminium plates in cooling. Error bars denote the standard deviation over five experiments. **e** Comparison of the cooling rates in the initial 8 s. Insets are schematics of the cooling mechanisms. Error bars denote the standard deviation over five experiments.

can levitate the droplet, insulating liquid/solid heat transfer[44,45]. To verify the efficiency of the liquid overlayer for superheated surfaces, we compare the cooling rates of our strategy with other liquid deposition enhancement methods for superheated surfaces. The same amount of water droplets (2 ml) works as a coolant to cool aluminium plates at 251 °C (10 × 10 × 0.25 cm). It is observed that the cooling rate using the pure water and polyethylene glycol (PEO) aqueous solution (1 wt%) is on par with that of natural cooling. Unlike such negligible effects, the overlayer strategy can bring about a fourfold improvement in the cooling rate (Fig. 4d, e). The energy conservation efficiency, defined as the ratio of the cooling heat to the heat capacity of water, is as high as 43.3% (Supplementary Note 2). Such a strategy opens an avenue to enhance water cooling.

**Post-deposition liquid control**. It is observed that by tuning the volume of the overlayer, the coated droplet can either be pinned or slide on a tilted surface, as though the support can be either fixed or roller type. We quantify the static friction $F_d$, the force required to initiate the motion of a static droplet, on surfaces of

contrasting asperities to establish a criterion of droplet state transition. A nanoporous surface with a structural thickness of ~300 nm and a micro-post array (height of 20 μm) decorated with nanoporosity are used as the testing surfaces (Fig. 5a), with static friction detected using a cantilever force sensor (Supplementary Fig. 9a, b)[16,46,47]. We define a length scale $V_o/D^2$, where $V_o$ is the volume of the overlayer and $D$ is the diameter of the droplet on the surface, as a rough equivalence to the lubricant thickness beneath the deposited droplet. As shown in Fig. 5b, on the nanotextured surface, the depinning force plummets from 173 to ~10 μN when $V_o/D^2$ increases from 0.459 to 3.7 μm. A similar trend also occurs on micro-/nanostructured surfaces with a mild transition. As $V_o/D^2$ reaches 23 μm, the depinning force decreases to ~10 μN, suggesting that the state transition occurs when $V_o/D^2$ surpasses a threshold whose magnitude relates to the structural height. Such an initial state transition occurs when the lubricant can immerse the asperities to reduce the liquid/solid contacts (Fig. 5c)[48]. When the overlayer volume is sufficient, deposited droplets freely slide until the depleted lubricant cannot sustain lubrication (Supplementary Note 3 and Supplementary Movie 7).

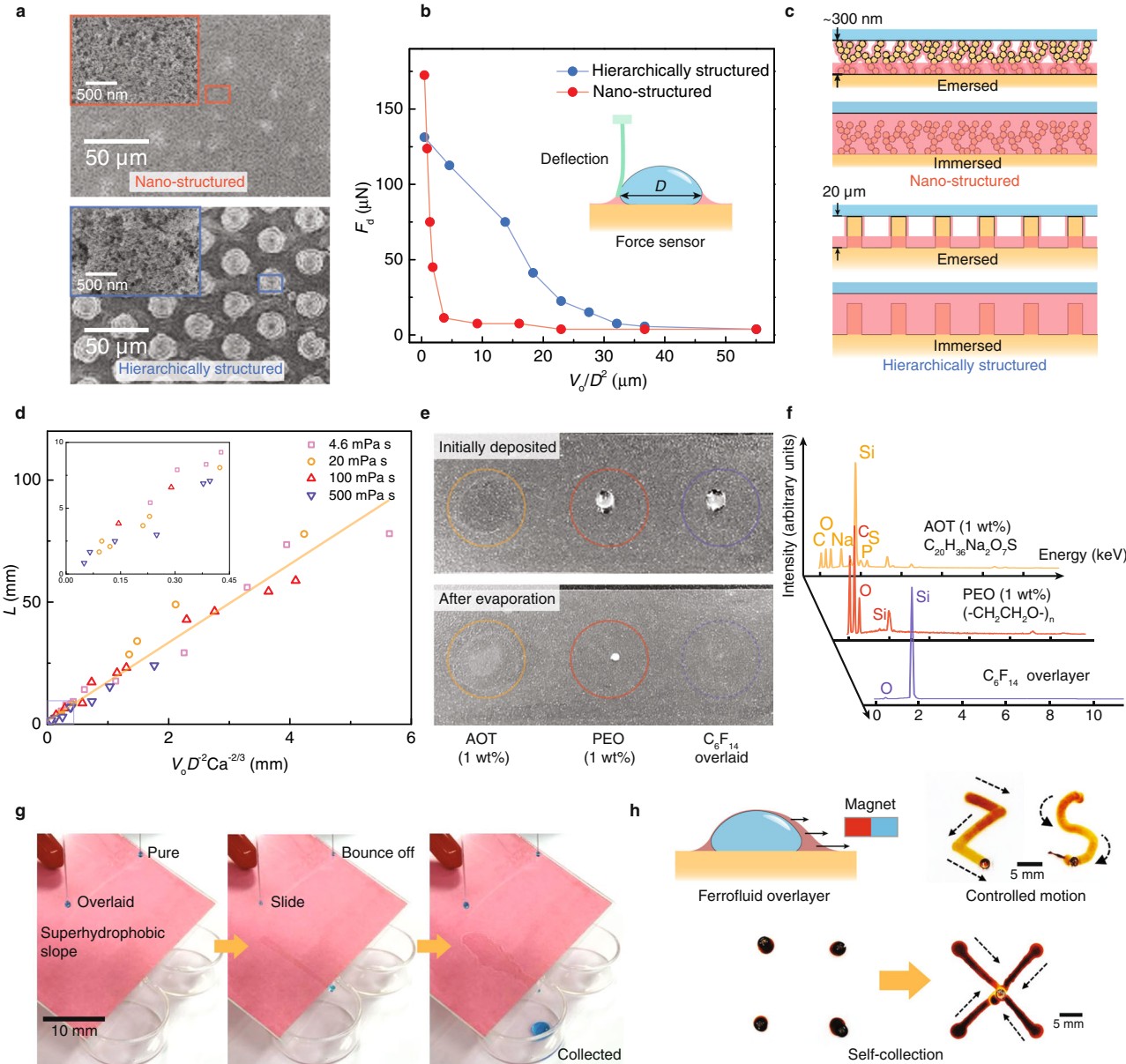

**Fig. 5 Post-deposition liquid control. a** SEM images showing the asperities on the nano-/hierarchically textured surfaces. **b** The depinning force as a function of the equivalent subjacent lubricant thickness $V_oD^{-2}$. Insets are schematics showing the detection of depinning forces. **c** Schematics showing the criterion for the state transition between pinned and sliding surfaces in (**a**). **d** The data collapse into a single curve for different systematic parameters, showing a linear relationship between the sliding distance and $V_oD^{-2}Ca^{-2/3}$. The orange line is the linear fitting line with a slope of 16.0. **e** Images showing the residues left by different deposition enhancing strategies. Compared with white stains using polymer additives, the volatile overlayer leaves no footprint. **f** Energy-dispersive spectroscopy confirms the composition of the polymer residues. **g** Demonstration of the liquid collection below a superhydrophobic slope. The overlayer suppresses the out-of-plane bouncing but enables in-plane sliding of water droplets. **h** Schematic and demonstrations showing the active control of overlaid droplets.

As shown in Fig. 5d, we then measured the maximum sliding length $L$, defined as the distance travelled by the droplet until $F_d$ increased on the nanotextured substrate by varying the systematic parameters, including $\mu_o$ (4.6–500 mPa s), $U_s$ (0.2–20 mm s$^{-1}$), $V_o$ (0.1–1.2 μl) and $D$ (2.7–10.1 mm). In agreement with the theoretical prediction, $L$ appears to be proportional to $V_oD^{-2}Ca^{-2/3}$ (Supplementary Note 3). We then traced the lubricant trail behind the sliding droplet by dying the lubricant black. The lubricant thickness can be roughly signified through the grey value. As shown in Supplementary Fig. 9g and h, in the region of $x \leq L$ the lubricant trail is relatively homogeneous, as predicted by the derivation (Supplementary Note 3). The thickness of the

homogeneous trail appears to be inversely related to the sliding velocity $U_s$ (Supplementary Fig. 9i). When a droplet travels beyond $L$, the lubricant is insufficient to sustain the lubrication, and the trail becomes inhomogeneous as the standard deviation of the grey value is larger for $x > L$ than for $x \leq L$ (Supplementary Fig. 9j).

We further demonstrate reliable liquid collection driven by gravity in Fig. 5g (Supplementary Movie 8). On a super-hydrophobic slope, dripping pure water droplets simply bounce away from the collecting petri dish. The addition of an overlayer eliminates out-of-plane bouncing and sustains in-plane mobility. In this way, dripping overlaid droplets deposit and slide on the

superhydrophobic slope and are collected in the dish. Apart from acting as the lubricant, the overlayer can be a force mediator by making it an external field-responsive. As shown in Fig. 5h, using ferrofluid as the overlayer we can actively control the in-plane motions through a single magnet (Supplementary Movie 9). Various trajectories can be readily defined and controlled in real time. As a magnet is centred, deposited droplets surrounding it can be self-directed towards the centre, suggesting a simplified paradigm to reliably capture and then automatically convey droplets on nonwetting surfaces.

For most applications, such as agriculture and cosmetics, minimizing the footprints of deposition enhancing methods is also crucial. Otherwise, the byproducts or residues left after evaporation can contaminate or toxify the surfaces. Figure 5e compares the residues with different deposition enhancing methods. The methods involving soluble polymer additives (1 wt% PEO and 1 wt% (sodium bis(2-ethylhexyl) sulfosuccinate, AOT)) were used for comparisons. Volatile liquids, perfluorohexane ($C_6F_{14}$), are used as the overlayer to eliminate the footprints without a performance trade-off. After evaporation, the PEO and AOT methods leave obvious white stains, whose compositions are confirmed using energy-dispersive spectrometry (EDS) detection (Fig. 5f). In stark contrast, no stains remained after the $C_6F_{14}$-overlaid droplet evaporated. EDS detects no fluorine signal on the deposited region. Thus, using volatile liquid as overlayers, our method can be made minimally invasive to the deposition surfaces.

## Discussion

By overlaying a tiny amount of lubricant (<0.1 vol% of the droplet) on the droplets, their interfacial properties are substantially modified. The behaviour of the overlaid droplets on nonwetting surfaces is similar to a mass-spring system attached with a damper-roller support, which is an unrecognized configuration that inhibits out-of-planes rebound while sustaining the in-plane mobility. The overlayer counteracts different surface terrains, spontaneously establishing liquid/solid contacts and draining kinetic energy through viscous dissipation. By increasing the lubricant volume, deposited droplets undergo state transitions from immobilized to sliding droplets, offering avenues for passive and active post-deposition liquid controls. The reconcilement of two seemingly contradictory mobilities has a wide range of applications, such as droplet-based printing, hazardous fluid handling and enhanced liquid/solid heat transfer.

## Methods

**Fabrication of the superhydrophobic surface**. A commercial spray, Glaco (Soft99), was used to coat the silicon wafer. The surface was then baked at 80 °C for 30 min to complete the coating.

**Fabrication of superomniphobic surface**. The suspension of polysiloxane/silica was prepared through hydrolytic condensation of 1H,1H,2H,2H-perfluorodecyltrichlorosilane (PFDTS) and tetraethyl (TEOS) in the presence of silica nanoparticles. First, silica nanoparticles (0.1 g) were dispersed in a solution containing 44 ml of anhydrous ethanol and 6 ml of ammonia aqueous solution. The mixture was ultrasonicated for 30 min. Then, 150 μl of PFDTS and TEOS were injected with vigorous stirring at 600 rpm. After reacting at room conditions for 24 h, a polysiloxane/silica suspension was formed. By spray coating the suspension (10 ml) onto vertically placed glass slides using an airbrush (Paasche H-SET) with 0.2 MPa $N_2$, the superomniphobic surface was prepared.

**Fabrication of micro-post array with nanostructures**. A layer of photoresist was coated on a silicon wafer ($P < 100 >$) at 1000 rpm for 45 s and then exposed with a mask aligner (MA 6, SUSS Micro Tec). After development, the silicon was etched by inductivity coupled plasma (ICP, GSE200, NAURA) with $O_2$ (36 sccm) and $SF_4$ (84 sccm). Then, commercialized spray Glaco (Soft99) was used to coat it with silanized silica nanoparticles.

**Overlaying droplets**. The droplets were overlaid using a co-flow microfluidics device. Two independent syringe pumps were used to precisely control the flow rates of the two phases.

**Characterization**. Fluid dynamics were visualized using a high-speed camera (Phantom M110) coupled with a camera lens (Sigma, 30 mm, f/1.4). The microscale structures were characterized using scanning electron microscopy (Hitachi S4800). The real-time temperatures of the aluminium plate were measured using thermocouples (K type), and the data were collected using a data acquisition unit (Keysight 34970A). The surface tension and interfacial tension of liquids were measured using a surface tensiometer (Cole-Parmer).

**Numerical study**. The three-dimensional numerical simulation was implemented using the laminar flow model. The multiphase Interfoam solver on the platform of OpenFOAM was used to solve the equations of continuity and momentum through a pressure-based solver. The volume of the fluid model was applied to show the interfaces of multiple phases (air, overlayer and droplet). The methodology PIMPLE algorithm, a combination of the PISO (pressure implicit slit operator) and SIMPLE (semi-implicit method for pressure-linked equations) algorithms, was implemented to simultaneously solve the mass and momentum equations. The computational domain is a $5D_0 \times 5D_0 \times 5D_0$ cube, consisting of more than 20 million cells.

## Data availability

The data that support the findings of this study are available from the corresponding authors upon reasonable request.

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

## Acknowledgements

The financial support from the Research Grants Council of Hong Kong (GRF17204420, 17210319, 17204718, 17237316 and CRF C1006-20WF and C1018-17G) is gratefully acknowledged. This work was also supported in part by the Zhejiang Provincial, Hangzhou Municipal and Lin'an County Governments.

## Author contributions

X.H., X.T. and L.W. designed the project. X.H. performed the experiments. X.H. and W. L. prepared the materials. H.Z., X.H. and X.T. planned and performed the simulations. J. L. provided constructive suggestions. X.H., X.T. and L.W. analysed the data. X.H., X.T. and L.W. wrote the manuscript. L.W. supervised the study. All authors commented on the paper.

## Competing interests

The authors declare no competing interests.
