## [Peer Review File · Nature Communications]

REVIEWER COMMENTS

Reviewer #1 (Remarks to the Author):

The authors present an experimental study on the impact of compound drops (water-in-oil and hexadecane-in-oil) on non-wetting surfaces. The paper focuses specifically on the drop rebound suppression after drop impact on non-wetting surface, in presence of an oil layer covering the core drop surface, i.e. water or hexadecane.

The paper is well written and structured, and the topic is relevant for the journal community, since compound drops represent a clear emerging trend.

Nonetheless, I noticed that the authors are missing the most recent references on the topic. The three main studies, listed here below, have initiated the study of compound drop impact, focusing on the core-shell drop configuration:

- Liu, D., & Tran, T. (2019). The ejecting lamella of impacting compound droplets. *Applied Physics Letters*, 115(7), 073702. <https://doi.org/10.1063/1.5097370>
- Liu, D., & Tran, T. (2018). Emergence of two lamellas during impact of compound droplets. *Applied Physics Letters*, 112(20), 203702. <https://doi.org/10.1063/1.5026821>
- Blanken, N., Saleem, M. S., Antonini, C., & Thoraval, M.-J. (2020). Rebound of self-lubricating compound drops. *Science Advances*, 6(11), eaay3499. <https://doi.org/10.1126/sciadv.aay3499>

Indeed, the authors have not considered publication from 2020 and only one from 2019, and the above works have been overlooked.

These publications need to be cited and thoroughly discussed, for the reasons detailed here below.

1. Blanken et al. focused exactly on the mechanism of drop rebound, describing how an oil layer can PROMOTE drop rebound on a smooth hydrophilic and hydrophobic surface, instead of SUPPRESSING it, as reported in this study for a superhydrophobic surface. The results appear, at first sight, contradictory. However, they offer the opportunity to better understand the mechanism of drop rebound. Clearly, the authors should discuss why the opposite phenomena are observed and make a comparative discussion. In particular, two questions arise: is suppression given by a different viscosity of the layer? Is roughness causing a breakup of the lubricating oil layer, so that rebound occurs on a smooth surface, and not on a rough surface? I think the key role is the breakup of the oil film, which roughness may cause. Still, on a smooth hydrophobic surface rebound occurs even when the oil layer breaks up, differently from what reported by the authors for superhydrophobic surfaces. All these questions and issues need to be properly addressed in the revised version of the manuscript.

2. "The critical value is the same for pure and overlaid droplets, an evidence showing that thin

overlayers have negligible impacts on the fluid dynamics except for the out-of-plane rebounding suppressions". This statement is too generic and essentially incorrect. Splashing, a relevant mechanism upon drop impact, is strongly affected by the presence of an oil layer. Spreading, as well, is affected. The authors should thus highlight the contribution of Tran et al. exactly on these two points, discussing them in the text.

Other minor comments:

- Figure 2c, y-axis. The scale should be changed, to appreciate if 0.25 is the correct slope.

Reviewer #2 (Remarks to the Author):

This manuscript studies the dynamics of a droplet coated by a layer of another immiscible liquid. The authors demonstrate that the coating can suppress rebound of the droplet from a non-wetting surface. On the other hand, the presence of the coating fluid around the droplet can enhance its mobility on the surface. The manuscript appears as a list of observations without much analysis of the physics observed. Some of these observations could have been interesting if a systematic study had been conducted, together with clear physical modelling. However, the authors only provide isolated observations and hypothesized explanations not supported by quantitative data. I therefore do not recommend publication of this manuscript.

Detailed remarks:

1. It would have been interesting for example to explore the range of parameters for which the addition of the coating layer suppresses bounding, changing systematically the impact velocity, the layer thickness or the coating liquid viscosity.
2. The discussions on the velocity gradient are not supported by any quantitative data, or modelling arguments.
3. The numerical simulations have quite low resolution. At this refinement level, how many cells capture the oil film?
4. It is not clear which liquid density is used in the definition of the Weber number on line 108.
5. Line 184: "Expectedly, L is proportional to the V_o/D^2 ": I do not see why there should be such proportionality relationship?
6. Lines 198-199: "the magnetic controlled motion leaves a more homogeneous lubricant trails (Fig. 4i)": This observation is representative of the whole manuscript: no quantitative data, no attempt to give any physical interpretation of the observation and no systematic variation of the control parameters.
7. Many figures are missing key information on the data presented to understand it: Weber number, coating layer thickness, drop size, ...
8. Many affirmations are not supported by quantitative data, such as the statement on the coating thickness on lines 80 to 82.
9. Line 178: The authors make a claim about a transition at 500 nm, while the figure cited (Fig. 4c) suggest a transition at 5 μm , so 10 times larger.

Response to the Comments by Referee #1

Comment 1: The authors present an experimental study on the impact of compound drops (water-in-oil and hexadecane-in-oil) on non-wetting surfaces. The paper focuses specifically on the drop rebound suppression after drop impact on non-wetting surface, in presence of an oil layer covering the core drop surface, i.e. water or hexadecane. The paper is well written and structured, and the topic is relevant for the journal community, since compound drops represent a clear emerging trend.

Answer: We thank gratefully the reviewer for his/her appreciation of this work.

Comment 2: Nonetheless, I noticed that the authors are missing the most recent references on the topic. The three main studies, listed here below, have initiated the study of compound drop impact, focusing on the core-shell drop configuration:

1. Liu, D., & Tran, T. (2019). The ejecting lamella of impacting compound droplets. *Applied Physics Letters*, 115(7), 073702. <https://doi.org/10.1063/1.5097370>.
2. Liu, D., & Tran, T. (2018). Emergence of two lamellas during impact of compound droplets. *Applied Physics Letters*, 112(20), 203702. <https://doi.org/10.1063/1.5026821>.
3. Blanken, N., Saleem, M. S., Antonini, C., & Thoraval, M.-J. (2020). Rebound of self-lubricating compound drops. *Science Advances*, 6(11), eaay3499. <https://doi.org/10.1126/sciadv.aay3499>.

Indeed, the authors have not considered publication from 2020 and only one from 2019, and the above works have been overlooked.

Answer: In accordance with the reviewer's comment 2, we have added work of Liu D. *et al.* (2019) as Ref. 35, work of Liu D. *et al.* (2018) as Ref. 36, and work of Blanken N. *et al.* as Ref. 37.

Comment 3: These publications need to be cited and thoroughly discussed, for the reasons detailed here below. Blanken *et al.* focused exactly on the mechanism of drop rebound, describing how an oil layer can PROMOTE drop rebound on a smooth hydrophilic and hydrophobic surface, instead of SUPPRESING it, as reported in this study for a superhydrophobic surface. The results appear, at first sight, contradictory. However, they offer the opportunity to better understand the mechanism of drop rebound. Clearly, the authors should discuss why the opposite phenomena are observed and make a comparative discussion. In particular, two questions arise: is suppression given by a different viscosity of the layer?

Answer: In accordance with the reviewer's comment 3, we have added context in Paragraph 3 (Page 5, Lines 114-119) and Fig. S3 to make a comparative discussion on the opposite phenomena and highlight that the suppression is given by the high overlayer viscosity.

Comment 4: Is roughness causing a breakup of the lubricating oil layer, so that rebound occurs on a smooth surface, and not on a rough surface? I think the key role is the breakup of the oil film, which roughness may cause. Still, on a smooth hydrophobic surface rebound occurs even when the oil layer breaks up, differently from what reported by the authors for superhydrophobic surfaces. All these questions and issues need to be properly addressed in the revised version of the manuscript.

Answer: In accordance with the reviewer's comment 4, we have added context in Paragraph 3 (Page 5, Lines 120-122) and Paragraph 1 (Page 6, Lines 123-129) and Fig. S4 to examine whether roughness cause a breakup of lubricating oil layer and discuss the roles of oil overlayer.

Comment 5: "The critical value is the same for pure and overlaid droplets, an evidence showing that thin overlayers have negligible impacts on the fluid dynamics except for the out-of-plane rebounding suppressions". This statement is too generic and essentially incorrect. Splashing, a relevant mechanism upon drop impact, is strongly affected by the presence of an oil layer. Spreading, as well, is affected. The authors should thus highlight the contribution of Tran *et al.* exactly on these two points, discussing the in the text.

Answer: In accordance with the reviewer's comment 5, we have deleted the statement of "The critical value is the same for pure and overlaid droplets, an evidence showing that thin overlayers have negligible impacts on the fluid dynamics except for the out-of-plane rebounding suppressions" and added context in Paragraph 2 (Page 6, Lines 130-136) and Paragraph 1 (Page 7, Lines 149-150) to discuss the influence of oil overlayer and highlight the contribution of Tran *et al.* on these two points.

Comment 6: Figure 2c, y-axis. The scale should be changed, to appreciate if 0.25 is the correct slope.

Answer: In accordance with the reviewer's comment 6, we have changed the scale of y-axis in Fig. 2c and added context in Paragraph 2 (Page 6, Lines 130-134) to confirm the slope.

We wish to take this opportunity to thank the Referee for his/her critical review and constructive comments/suggestions.

Response to the Comments by Referee #2

Comment 1: This manuscript studies the dynamics of a droplet coated by a layer of another immiscible liquid. The authors demonstrate that the coating can suppress rebound of the droplet from a non-wetting surface. On the other hand, the presence of the coating fluid around the droplet can enhance its mobility on the surface. The manuscript appears as a list of observations without much analysis of the physics observed. Some of these observations could have been interesting if a systematic study had been conducted, together with clear physical modelling. However, the authors only provide isolated observations and hypothesized explanations not supported by quantitative data. I therefore do not recommend publication of this manuscript.

Detailed remarks:

Answer: We thank gratefully the reviewer for his/her appreciation of observations in this work.

Comment 2: It would have been interesting for example to explore the range of parameters for which the addition of the coating layer suppresses bounding, changing systematically the impact velocity, the layer thickness or the coating liquid viscosity.

Answer: In accordance with the reviewer's comment 2, we have added context in Paragraph 3 (Page 6, Lines 146-147), Paragraph 1 (Page 7, Lines 148-151), and Fig. 2f, S6 to explore the range of parameters for which the addition of the coating layer suppresses bounding by changing systematically the impact velocity, overlayer thickness, and viscosity.

Comment 3: The discussions on the velocity gradient are not supported by any quantitative data, or modelling arguments.

Answer: In accordance with the reviewer's comment 3, we have added context in Paragraph 2 (Page 7, Lines 154-167), Fig. 3a, S8 to replace the discussion on the velocity gradient with quantitative distribution of viscous dissipation rate.

Comment 4: The numerical simulations have quite low resolution. At this refinement level, how many cells capture the oil film?

Answer: In accordance with the reviewer's comment 4, we have fined the numerical simulations and added context in Supplementary Note 1 (Page S12, Lines 86-87) to provide the cell numbers.

Comment 5: It is not clear which liquid density is used in the definition of the Weber number

on line 108.

Answer: In accordance with the reviewer's comment 5, we have added context in Paragraph 2 (Page 5, Lines 107-110) to clarify the liquid density used in the definition of Weber number.

Comment 6: "Expectedly, L is proportional to the V_0/D^2 ": I do not see why there should be such proportionality relationship?

Answer: In accordance with the reviewer's comment 6, we have added context in Paragraph 1 (Page 9, Lines 211-217), Fig. 5d, S9, and Supplementary Note 3 to establish a proper relationship between the sliding length and systematic parameters.

Comment 7: Lines 198-199: "the magnetic controlled motion leaves a more homogeneous lubricant trails (Fig. 4i)": This observation is representative of the whole manuscript: no quantitative data, no attempt to give any physical interpretation of the observation and no systematic variation of the control parameters.

Answer: In accordance with the reviewer's comment 7, we have added context in Paragraph 1 (Page 9, Lines 217-221) and Paragraph 1 (Page 10, Lines 222-223), Fig. S9g-j, and Supplementary Note 3 to provide quantitative data with physical interpretation and variation of control parameters.

Comment 8: Many figures are missing key information on the data presented to understand it: Weber number, coating layer thickness, drop size, ...

Answer: In accordance with the reviewer's comment 8, we have added key information (Weber number, coating layer thickness, coating layer viscosity, droplet size, and impact velocity) in Fig. 1d-f, 3, 4a-b, and their captions and the caption of Fig. 2.

Comment 9: Many affirmations are not supported by quantitative data, such as the statement on the coating thickness on lines 80 to 82.

Answer: In accordance with the reviewer's comment 9, we have added context in Paragraph 1 (Page 4, Lines 77-82) to provide quantitative description for the statement on the coating thickness.

Comment 10: Line 178: The authors make a claim about a transition at 500 nm, while the figure cited (Fig. 4c) suggest a transition at 5 μm , so 10 times larger.

Answer: In accordance with the reviewer's comment 10, we have added context in Paragraph

1 (Page 9, Lines 205-206 and 208-209) to re-examine the claim by improving the experiment in Fig. 5b.

We wish to take this opportunity to thank the Referee for his/her critical review and constructive comments/suggestions.

Response to the Comments by Referee #1

Comment 1: The authors have addressed my comments and the issues I had raised. As such, my suggestion is now to publish the paper.

Answer: We thank gratefully the referee for his/her appreciation of this work and our revision.

Response to the Comments by Referee #2

Comment 1: The authors have significantly improved the manuscript by adding more systematic data. The overall quality of the manuscript is now better, making significant contributions on the manipulation of droplets.

Answer: We thank gratefully the referee for his/her appreciation of this work and our revision.

Comment 2: The data presented in Fig. 2(f) and Fig. S6 especially is very interesting, although not understood. I recommend adding a reference to Fig. S6 in the caption of Fig. 2(f). Why is the data for 500 mPa s from Fig. S6(a) not presented as solid line?

Answer: In accordance with the referee's comment 2, we have added a reference to Supplementary Fig. 6 in the caption of Fig. 2(f) (Page 20, Lines 1-2). The data for 500 mPa s from Supplementary Fig. 6 has also been added in Fig. 2(f) as a purple solid line.

Comment 3: I just have a minor remark that the authors can consider before publication: I am still not convinced by the scaling of the “work of the lubricant pinning” presented on line 128. The discussion on the “depinning force” could probably be improved by discussing the following recent paper: Keiser, A., Baumli, P., Vollmer, D., & Quéré, D. (2020). Universality of friction laws on liquid-infused materials. *Physical Review Fluids*, 5(1), 014005. <https://doi.org/10.1103/PhysRevFluids.5.014005>

Answer: In accordance with the referee's comment 3, we have revised context in Page 6, Lines 8-10 to explain the work of the lubricant pinning. The mentioned literature about the friction forces which causes viscous dissipation has also been cited as Ref. 38 (Page 6, Line 8).

We wish to take this opportunity to thank the referee for his/her critical review and constructive comments/suggestions.

REVIEWERS' COMMENTS

Reviewer #1 (Remarks to the Author):

The authors have addressed my comments and the issues I had raised. As such, my suggestion is now to publish the paper.

Reviewer #2 (Remarks to the Author):

The authors have significantly improved the manuscript by adding more systematic data. The overall quality of the manuscript is now better, making significant contributions on the manipulation of droplets. The data presented in Fig. 2(f) and Fig. S6 especially is very interesting, although not understood. I recommend adding a reference to Fig. S6 in the caption of Fig. 2(f). Why is the data for 500 mPa.s from Fig. S6(a) not presented as solid line?

I just have a minor remark that the authors can consider before publication: I am still not convinced by the scaling of the "work of the lubricant pinning" presented on line 128. The discussion on the "depinning force" could probably be improved by discussing the following recent paper:

Keiser, A., Baumli, P., Vollmer, D., & Quéré, D. (2020). Universality of friction laws on liquid-infused materials. *Physical Review Fluids*, 5(1), 014005.
<https://doi.org/10.1103/PhysRevFluids.5.014005>